# SiO_2_-PVA-Fe(acac)_3_ Hybrid Based Superparamagnetic Nanocomposites for Nanomedicine: Morpho-textural Evaluation and *In Vitro* Cytotoxicity Assay

**DOI:** 10.3390/molecules25030653

**Published:** 2020-02-04

**Authors:** Ana-Maria Putz, Cătălin Ianăși, Zoltán Dudás, Dorina Coricovac, Claudia (Farcas) Watz, Adél Len, László Almásy, Liviu Sacarescu, Cristina Dehelean

**Affiliations:** 1”Coriolan Dragulescu” Institute of Chemistry, Romanian Academy, Mihai Viteazul Bd., No. 24, 300223 Timişoara, Romania; lacramamanamaria@yahoo.com (A.-M.P.); cianasic@yahoo.com (C.I.); 2Wigner Research Centre for Physics, POB 49 1525 Budapest, Hungary; 3Pharmacy II Department, Faculty of Pharmacy, “Victor Babes ¸” University of Medicine and Pharmacy, 2 Eftimie Murgu Sq., 300041 Timisoara, Romania; dorinacoricovac@umft.ro (D.C.);; 4Centre for Energy Research, Konkoly-Thege 29-33, 1121 Budapest, Hungary; len.adel@energia.mta.hu; 5University of Pécs, Faculty of Engineering and Information technology, Boszorkány St. 2, 7624 Pécs, Hungary; 6Institute of Macromolecular Chemistry “Petru Poni”, Aleea Grigore Ghica Voda, nr. 41A 700487 Iasi, Romania; sacarescu@yahoo.com

**Keywords:** magnetic silica hybrids, maghemite, morpho-textural characterization, toxicology, sol-gel technique

## Abstract

A facile sol-gel route has been applied to synthesize hybrid silica-PVA-iron oxide nanocomposite materials. A step-by-step calcination (processing temperatures up to 400 °C) was applied in order to oxidize the organics together with the iron precursor. Transmission electron microscopy, X-ray diffraction, small angle neutron scattering, and nitrogen porosimetry were used to determine the temperature-induced morpho-textural modifications. In vitro cytotoxicity assay was conducted by monitoring the cell viability by the means of MTT assay to qualify the materials as MRI contrast agents or as drug carriers. Two cell lines were considered: the HaCaT (human keratinocyte cell line) and the A375 tumour cell line of human melanoma. Five concentrations of 10 µg/mL, 30 µg/mL, 50 µg/mL, 100 µg/mL, and 200 µg/mL were tested, while using DMSO (dimethylsulfoxid) and PBS (phosphate saline buffer) as solvents. The HaCaT and A375 cell lines were exposed to the prepared agent suspensions for 24 h. In the case of DMSO (dimethyl sulfoxide) suspensions, the effect on human keratinocytes migration and proliferation were also evaluated. The results indicate that only the concentrations of 100 μg/mL and 200 μg/mL of the nanocomposite in DMSO induced a slight decrease in the HaCaT cell viability. The PBS based *in vitro* assay showed that the nanocomposite did not present toxicity on the HaCaT cells, even at high doses (200 μg/mL agent).

## 1. Introduction

In the past years great efforts have been dedicated to synthesizing superparamagnetic iron oxide nanoparticles, as promising materials for various applications: contrast agent in magnetic resonance imaging (MRI) [1], biocatalyst [2], biomarker [3] and biosensor [4], carrier for drugs [5], and nanomedicine for magnetic hyperthermia treatment [6]. For biomedical applications the nanoparticles are usually prepared in colloidal medium [7], where the absence of a specific coating leads to the formation of aggregates and, finally, to agglomeration, with the latter being an irreversible process [8]. Various synthetic processing methods have been developed to avoid the agglomeration [9,10], with the most promising seeming to be the incorporation in silica matrices [11]. Silica supports offer chemical stability, biocompatibility, easily tunable structure, and easy functionalization [12,13]. To be accepted for bio-related applications, these materials need to possess specific features, such as: nanoscale sizes, proper shape, biocompatibility, lack of toxicity, and biodegradation [14]. Silica carriers, assuring the ability of material to accomplish its envisaged job, fulfill these requirements. 

Besides the defined requirements for biocompatibility, the physicochemical properties, like particle size and shape, porosity, specific surface area, and superparamagnetic properties, of the nanomaterials play an important role in their applicability [15]. Specifically, the physicochemical properties and functionalities can be tailored by rigorous control of the synthesis parameters [16,17,18,19]. There are several physicochemical characterization methods that offer a complete picture about the properties of the magnetic nanoparticles containing samples; the most suitable for morpho-textural characterizations are: nitrogen porosimetry, electron microscopy, and small angle neutron scattering. Those specific methods have been used to obtain information about the pore and particle size (in the domain of 1–100 nm) of the magnetic nanoparticles [20,21,22].

In the case of nanomaterials, the control of morphology and particle size is extremely important, as particle size and shape often dictate the magnetic behaviour [23] and their suitability in various application domains. Following the particle size criterion, Singh presented the main categories of superparamagnetic materials that are suitable for usage inside the human body [24]. Daniel Esqué-de los Ojos described how the pore size influences the properties of materials [25].

The *in vitro* cytotoxicity has to be assessed before the qualification of the nanomaterials for the *in vivo* testing [16,17]. Recently, the concept of “cell type dependent nanotoxicity” was introduced and conferred an original view upon the certain aspects that need to be evaluated before the introduction in the human organism [26]. 

Among others, Yu et al. presented a pertinent study dealing with the cytotoxicity of bare and coated magnetic materials [27]. According to this study, even high concentrations of magnetic material do not significantly reduce the cell viability, neither that of malignant lung tumour A549 cell line nor of the HeLa human cells. It was observed that the toxicity was reduced when the bare magnetic material was coated with dextran, silica, and polyethylene glycol [27]. 

Similar studies on different cell lines showed that if the magnetic materials are covered, they did not present toxicity, even when the concentrations were high [18,28,29,30,31].

Among the inorganic magnetic nanomaterials used more frequently to interact with human organism, metal oxides, especially iron oxides (magnetite, maghemite), silica, titanium oxides, or zinc oxide are found [32]. These oxides, even when they are covered and present no cytotoxicity and before they could harm the living bodies as toxic elements by reaching a certain level of concentration, they could mechanically harm the organs and tissue of the body due to their inappropriate size or due to sharp feature of geometric shape. At the same time, the body fluids, due to their thermal or chemical relative aggressivity, could oxidize or decompose the magnetic material. As a consequence, the magnetic particles and the living organism both need to be protected. Coating with inert or active mono or multilayers offers a solution, additionally improving the stability and even the functionalities of the magnetic nanoparticles. Elias et al. [33] showed that surface modification is necessary in order to prevent agglomeration, to reduce the toxicity, and allow for the optimal attachment of therapeutics particle. Properly choosing the synthesis reactants and controlling synthesis parameters can accomplish this [33,34]. Baber et al. [35] found that, by embedding the magnetic material in amorphous silica, its chemical stability significantly increased, and, due to the changes in soluble iron concentration and/or oxidative stress, the cellular responses decreased [35]. Malvindi, et al. [36] showed that, by introducing organic polymers in magnetic materials, the hydrophilic proprieties could be improved, thus making them suitable for medical applications, especially in MRI [36]. 

The biocompatibility of various silica coated-magnetic nanoparticles has already been demonstrated on multiple *in vitro* models. Foglia and co-workers [37] revealed that silica-coated magnetic iron oxide nanoparticles with diameter under 5 nm are biocompatible structures and do not interfere with the cell cycle distribution or with the expression of key differentiation markers and do not affect pro-inflammatory cytokines response [38]. Another study revealed that the silica-coated magnetite nanoparticles induced no toxicity in zebrafish, due to high protective feature of the silica layer to cellular biodegradation processes that impeded the direct exposure of magnetite nanoparticle to cells/tissues, which could lead to endocytosis and the accumulation of nanoparticles, inducing detoxification processes and cellular/tissue damage [38]. 

The biocompatibility of magnetic nanoparticles was evaluated on various cell types, among which (immortalized human keratinocytes) HaCaT cells and (human melanoma) A375 cells are commonly used [39,40,41,42]. 

HaCaT (human keratinocytes) were selected as the non-tumorigenic/normal cell model to assess the cytocompatibility of the magnetic nanocomposite sample based on the several following arguments: i) is an immortalized cell line, presenting the advantage of expressing constant characteristics that do not show DNA or chromosomal alterations under long-term cultivation conditions [43] ii) an *in vitro* experiment that was developed on HaCaT cells to test silica nanoparticles cytotoxicity indicated a dose-dependent cytotoxic effect characterized by reduced cell viability and damaged cell membrane integrity with internalization of the nanoparticles within cytoplasm, lysosomes, and autophagosomes [44]; iii) after intravenous administration of silica nanoparticles into mice, skin was one of the organs where the nanoparticles were detected along with muscle, bone and liver [45]; and, iv) these cells were also chosen as the normal cell model in several recent studies that evaluated the cytotoxicity of various compounds as MRI agents [46,47,48].

The A375 human melanoma cells were selected as the tumour cell model to evaluate the cytotoxicity of the material, because this cell line could be used for further *in vivo* evaluations that are intended to assess the *in vivo* MRI potential of test samples, in mice that were xenografted with A375 cells. Several studies have already used human epidermoid carcinoma/melanoma xenografts to evaluate the *in vivo* potential of diverse MRI test agents [49,50]. Moreover, Huang X. and collaborators assessed the effect of mesoporous silica nanoparticles on tumour proliferation while using a model of nude mice xenografted with A375 cells [51]. Additionally, the following reasons led us to the selection of this cell line: i) both superparamagnetic iron and silica nanoparticles proved to be reliable nanoplatforms for the detection and treatment of different cancers, including melanoma [40,52,53,54,55] and ii) A375 melanoma cell line presents the characteristics of the human genitor and possess B-RAF and CDKN2 mutations, which are typical of cutaneous melanoma, representing an eligible candidate for the development of *in vivo* models [56].

In the present study, the iron oxide silica nanocomposite samples with target iron oxide concentration of 20% Fe_2_O_3_ were obtained by the calcination of an inorganic-organic hybrid nanocomposite xerogel and were characterized by nitrogen porosimetry, small angle neutron scattering (SANS), and transmission electron microscopy (TEM). After the calcination at different temperatures, the 260 °C fired sample was selected to be tested for cytotoxicity. Among the thermally treated materials, this sample exhibited optimum magnetic properties, specifically the highest value of saturation magnetization, σ_sat_ = 48.1 emu/g, low coercivity (H_c_ = 0.021 kOe), and remanence of σ_r_ = 1.45 emu/g, giving a value of 0.03 for the σ_r_/σ_sat_ ratio [57], which allows us to assign superparamagnetic behaviour to the sample P-260. 

Promising results were obtained when this sample was tested as a contrast agent in medical imaging [57]. By supplementary characterization methods, it has been shown that these materials fully fulfill the requirements for this application, including the lack of toxicity. Therefore, the goal of this study was to qualify the materials as for their biomedical applicability, such as drug carriers, and demonstrate their nontoxicity and suitable structural properties. 

## 2. Results and Discussion

### 2.1. Nitrogen Porosimetry

The nitrogen sorption method characterized the hybrid silica-iron oxide xerogel thermally treated at different temperatures. Figure 1 presents the evolution of the isotherms and the morpho-textural parameters that were obtained from nitrogen adsorption-desorption isotherms are presented in Table 1. The sample descriptions are given in Section 3.1. The nanocomposites are named as P-x, where x is the temperature of thermal treatment.

The hybrid nanocomposite xerogel sample that was obtained at 120 °C had a compact nonporous structure, consisting of iron precursor and organics [58]. As a consequence, the adsorbed nitrogen amount was very small and did not allow for obtaining an isotherm plot. Increasing the thermal treatment temperature to 220 °C, a complex isotherm was obtained. Apparently being a combination of Type II reversible isotherm and Type IV(a) isotherm with H4 Hysteresis, it indicates an extended pore size distribution from microporous to mesoporous domain. 

At 260 °C, the specific surface area and total pore volume increased, because most of the organic phase and precursors decomposed [58]. Thus, the free volumes appear in the samples that were calcined above 260 °C. Therefore, the increases are due to the partial carbonization, oxidation, and volatilization of the organic parts (acetylacetonate and PVA). This isotherm is a combination of Type IV(a) isotherm (H4 hysteresis) and I(b) reversible isotherm, indicating a significant percentage of microporosity when comparing to mesoporosity. At 300 °C, all of the samples exhibited the maximum values of specific surface area, total pore, and micropores volume, which was consistent with literature data and also with our previous works [57,58]. At this temperature, the volatile products were eliminated in a large amount without a significant collapse of the silica pore walls. In case of the sample calcined at 300 °C, a combination of Ib and IVa isotherms was observed, with the H4 loop. The small hysteresis in the range ~0.75–1 P/Po indicates a certain amount of macropores [59] having pore sizes in the 50 nm–80 nm domain. The sample fired at 400 °C showed a reversible isotherm, along the entire P/Po range, of type Ib. The specific surface area and total pore volume was diminished when comparing to the 300 °C fired sample. Two opposite processes occurring simultaneously caused this, the removal of the volatile content of the pores and the pore walls thermal collapse. 

For samples that were obtained by calcination at 220 °C and 260 °C, the isotherms are not closed at all. However, for the sample obtained by calcination at 260 °C, a small hysteresis can be noticed. The closing was not perfect due to presence of organic part, but when the temperature has been increased the allure of sample changed, showing Type Ib isotherm.

The micropores surface area was estimated by using V-t plots. The type of hysteresis was deducted by further analysis. We used both models, the BJH (we used the adsorption branch because the desorption is not closing completely) and the DFT. While the BJH method gives the pore size distribution valid only for the mesoporous size range, the DFT method comprises the mesoporous and as well the microporous range, thus it might be more accurate. Comparatively, it can be observed that both of these results indicate the same tendency.

For the samples fired from 220 °C up to 400 °C, the total pore volume increased four times when comparing to the sample calcined at 120 °C, as can be seen in Table 1. 

The evolution of the microporosity/mesoporosity ratio with the thermal treatment temperature has a parabolic shape (Figure 2). The most microporous sample is the one obtained at 300 °C, where 75% percent of the porosity is in the microporous domain. The pore filling effect due to the crystal growth is followed by reduction of the specific surface area, and by the decreasing of the micropore percentage at 400 °C.

A continuous decrease of average pore diameter can be observed (Figure 3) while analysing the pore size distribution evolution between the samples that were obtained by calcination at different temperatures, which is consistent with the values that were obtained for the surface area data collected for the materials obtained up to 300 °C.

Specifically, the surface area increased by increasing the number of pores. Continuing to rise the calcination temperature up to 400 °C, the pore diameter slightly increased and, correspondingly, the surface area decreased.

Examining FHH data that were derived from the N_2_ adsorption-desorption isotherms, the fractal character of the material surface texture can be deduced, and the evolution of the surface roughness with calcination temperature can be followed. Similarly to the interpretation of Q. Wei and D. Wang [60], we concluded that the synthesized material was structured on the basis of dense primary particles and their aggregates with surface roughness. The surface fractal dimension dfs, as calculated with FHH method, as it was expected, was increasing with a firing temperature up to 300 °C (Table 1).

### 2.2. Thermal Analysis

Thermal analysis (TG/DTG/DTA) was carried out to determine the temperature of crystalline phase formation. The DTG/DTA curves revelead four different peaks (Figure 4). The first small and broad endothermic effect can be observed around 70 °C, and it can be attributed to the removal of the liquid fraction (solvents and water) superficially adsorbed in the silica pores.

The range of 125–225 °C is defined by an exothermic effect with a maximum around 190 °C (Table 2); here, the simulaneous melting and the fractional decomposition of the acetyl acetate-iron (III) occurs. At the same time, the loss of structural water and the elimination of organics take place.

The third thermal effect is also exothermic and takes place in the rage of 225–275 °C. This effect is caused by the elimination of organic moieties, progress of the silanol groups polycondensation, and fracture in polymers. At the same time, the transformation of the acetylacetonate groups into maghemite (γ-Fe_2_O_3)_ takes place. 

During the last exothermic effect, the elimination of the organic residues and final polycondesation of the silanol groups takes place. All of the acetylacetonate groups are transformed to γ-Fe_2_O_3_ and the formation of the hematite (α-Fe_2_O_3_) could be noticed. At 500 °C, all of the maghemite is transformed into hematite. The total weight loss is 51.44%.

### 2.3. Transmission Electron Microscopy

The evolution of the crystalline phase and the morphological changes of the silica matrix, induced by the thermal treatment temperature were visualized by TEM images (Figure 5). Amorphous silica phase was observed with some disordered crystalline spots of the iron precursor for the material obtained at 120 °C. In the case of materials obtained starting from 220 °C, the formation of the crystals is clearly visible as well their growths, with the increase of thermal treatment temperature. Up to 300 °C, only cubic maghemite phase could be determined, while, at 400 °C, the hematite phase appeared in addition. With the thermal treatment, the crystallite size is increasing, while the polycrystallinity is present at every thermal treatment temperature. The crystallite sizes are in good correlation with the sizes determined from the X-Ray diffraction (XRD) and SANS measurements.

### 2.4. Small Angle Neutron Scattering (SANS) 

The SANS curves were analysed by the analytical model that combines the scattering from the silica pore surface and the scattering from the nanosized metallic particles of arbitrary shapes, while using the Guinier approximation. The model fitting was performed using the Igor Pro software [61].

Equation (1) presents the Beaucage model [62], where Rg is the gyration radius of the iron oxide crystallites and p is a power law exponent related to the fractal dimension, A and B are constants, which depend on the number and volume of the scattering particles, and their neutron scattering length densities.
(1)$$I\left(Q\right)=A\;exp\left(\frac{-Q^{2}Rg^{2}}{3}\right)+B{\left[\frac{{\left(erf\frac{QRg}{\sqrt{6}}\right)}^{3}}{Q}\right]}^{p}$$

In Figure 6, the measured scattering data and the fitted model curves are shown, while Table 3 lists the values of the fitting parameters.

At 120 °C calcination temperature, the scattering curve displays a flat incoherent background at high Q values, which is due to the residual water and organics content of the materials. At low Q, a single power law is seen, which is characteristic of the porous xerogel structures [63,64,65]. There is no evidence on the formation of crystallites, due to their small quantity. 

At higher temperatures, the data show a distinct scattering feature of nanometer sized particles, namely the bending of the curves below Q = 0.03 Å^−1^. At 220 °C, the scattering signal is lower than at higher temperatures, which indicated the beginning of the formation of the crystallites. At this stage, iron oxide grains, organic pore fillers, and the unreacted metallic precursors are still present in the material. Starting from the calcination temperature of 260 °C, scattering from the iron-oxide crystallites was observed; the size of the crystallites increased with the temperature (Table 1). At the highest studied temperature, 400 °C, the average crystallite size was about 25 nm.

The fractal dimensions (see Equation (1)) that were obtained from the model fitting [66] characterize the pore surface, and the interface between the xerogel and the iron-oxide crystallites. At the lowest calcination temperature, the surface fractal character is explained by the presence of the organic pore fillers and the unreacted iron precursors that partly fill the xerogel pores. This is also confirmed by the N_2_ adsorption analysis. At higher temperatures, the scattering shows volume fractal behaviour. The volume fractal dimension decreases with the increase of the temperature, due to the presence of a progressive phase transition process of the iron-oxide, being accompanied by the growth of the crystallites [57].

The structural investigations showed that the sample that was calcined at 260 °C had the best magnetic and structural properties, and it was chosen for further toxicological evaluation for perspective biomedical applications.

### 2.5. Crystallite Size

The XRD diffractograms and the crystalline size determination were exhaustively presented in our previous article [57]. In Figure 7, the crystallite sizes determined by different methods are compared. The same evolution of the crystallite size was evidenced by TEM and XRD. The SANS data match perfectly with XRD at 260 °C and 300 °C, while at 400 °C an average crystallite size was obtained. 

### 2.6. Zeta Potential Measurements

Zeta potential, as a descriptor of the particles agglomeration, is based on the technique of dynamic light scattering in an electric field. Positive values, +242.13 mV and +41.32 mV, were recorded in chloroform and bi-distilled water, respectively. These values correspond to colloidal solutions without strong particle clustering; a range between −30 and +30 mV is usually considered [67,68] to indicate a strong tendency for agglomerations.

### 2.7. Toxicological Evaluation of Magnetic Nanocomposite P-260

The effect that was induced by nanocomposite P-260 (referred further as NC) sample was tested on two different cell lines, one cell line of human keratinocytes (HaCaT cells) and one tumour cell line of human melanoma cells (A375 cell line). The viability of the cells was assessed after exposure to five different concentrations (10 μg/mL, 30 μg/mL, 50 μg/mL, 100 μg/mL, and 200 μg/mL) of NC sample, for 24 h.

The samples were solubilized in both DMSO (dimethyl sulfoxide) and PBS (phosphate saline buffer), to evaluate the solvent dependence effect of HaCaT and A375 cells on the NC sample. Determinations were made at different DMSO content, and Figure 8, Figure 9, Figure 10 and Figure 11 present the obtained results.

The cells’ viability was compared to that of the control cells, unstimulated cells, considered to have 100% viability. Figure 8A shows that, at low concentrations, 10 μg/mL, 30 μg/mL, and 50 μg/mL, DMSO has a low stimulating effect on HaCaT cell viability, whereas at higher concentrations, of 100 μg/mL and 200 μg/mL, respectively, the cells manifested almost the same viability as the control cells.

The cell viability percentage that was obtained after treatment was normalized to the cell viability percentage obtained for the cells exposed to the same concentration of solvent (DMSO) to eliminate the effect induced by DMSO treatment. Figure 8B presents the apparent decrease of the normalized cell viability rate of the NC treated cells. It seems that only the highest tested concentration (200 μg/mL) induced an inhibitory effect on HaCaT cell population. All of the other concentrations (10 μg/mL, 30 μg/mL, 50 μg/mL, and 100 μg/mL) are considered to be non-toxic for healthy cells.

From Figure 9A can be observed that the viability of healthy cells was not affected by the treatment with PBS, even after exposure to the highest test concentration. Moreover, the results that are presented in Figure 9B indicate that the NC sample solubilized in PBS presents no cytotoxicity on HaCaT cell population, regardless of concentration.

The results that are presented in Figure 10A show that DMSO at the concentrations tested in this experiment did not induce a reduction of human melanoma cell viability, except for the concentration of 50 μg/mL, where a slight cell viability decrease could be observed. In this situation, the cells showed 88.74% cell viability.

Following stimulation of the human melanoma tumour cell line A375 with the test substance solubilized in DMSO (Figure 10B), a slight increase in cell viability at low concentrations (10 μg/mL, 30 μg/mL, and 50 μg/mL) was observed, relative to the same DMSO concentrations. At concentrations of 100 μg/mL and 200 μg/mL, a slight cytotoxic effect was detected by the reduction of the viable A375 cells number.

Figure 11A presents the effect induced by 24 h PBS-treatment on human melanoma A375 cell viability. At concentrations of 10 μg/mL, 30 μg/mL, and 50 μg/mL, a slight stimulatory activity was observed, while, at a concentration of 100 μg/mL and 200 μg/mL, the reduction in viability of tumour cells of approx. 10% was noticed. A few variations in the percentage of viable cells at concentrations of 100 μg/mL and 200 μg/mL were observed as a result of both stimulations with PBS and the test substance of the NC sample. This effect is similar to the one that was observed on samples that were solubilized in PBS.

On the basis of these results, we can recommend PBS as a solvent for *in vitro* test assays due to the lack of toxicity on healthy cells, even at high doses of 200 μg/mL, however the cytotoxic effect on tumour A375 cells proved to be similar to that observed in samples that were solubilized in DMSO.

### 2.8. Evaluation of P-260 (NC) Nanocomposite by Studying the Effect on Human Keratinocytes Migration and Proliferation Potential

A decrease of HaCaT viable population was observed after stimulation with NC that was dissolved in DMSO, therefore the effect of the sample on migration and proliferation of human keratinocytes was also verified, while employing the scratch assay technique. Stimulation was performed while using the same concentrations of NC in DMSO that were used for cytotoxicity testing and Figure 12 presents the results. 

It can be observed that the compound stimulates cell migration, with the scratched area being almost totally repopulated with cells, at low concentrations of 10 μg/mL and 30 μg/mL. At higher concentrations (50 μg/mL, 100 μg/mL, and 200 μg/mL), the effect is the opposite, namely the cells migration and traced line repopulation was inhibited. Analysing the results that were obtained for the cytotoxicity assessment and the ones obtained from the scratch assay, it was concluded that, in spite of small inhibitory activity on HaCaT viability at low concentrations (10 μg/mL, 30 μg/mL) of NC solubilized in DMSO, these concentrations of NC did not affect HaCaT migration and proliferation activity, if compared to control cells response.

*In vitro* assays are dedicated methods for evaluating the biocompatibility/cytotoxicity of different magnetic nanoparticles. As previously proven [69], the potential cytotoxic effect that is manifested by magnetic nanoparticles is dependent of the coating agent, particle size, porosity, shape, test cell line, and the stimulation time employed for the *in vitro* testing. The same study revealed that, besides the mentioned parameters, the effect of the used solvent needs to be taken into account, because it also influences the cytotoxicity of the test samples.

In this study, the biocompatibility of NC sample was also assessed on HaCaT and A375 cells. The obtained results showed that the healthy cell line (HaCaT) manifested good viability after exposure to all five concentrations (10 μg/mL, 30 μg/mL, 50 μg/mL, 100 μg/mL, and 200 μg/mL) of the P-260 sample solubilized in PBS. However, a slight decrease of HaCaT viability was observed when the cells were exposed to the highest concentration (200 μg/mL) of the P-260 sample solubilized in DMSO. These data sustained that the P-260 sample was a biocompatible compound at concentration up to 200 μg/mL, if PBS is used as a solvent. Moreover, high concentrations (100 μg/mL, 200 μg/mL) of the P-260 sample induced a slight decrease (approx. 10%) of human melanoma–A375 cells viability.

## 3. Materials and Methods

### 3.1. Materials and Synthesis Protocol

The starting materials were: iron(III) acetylacetonate [Fe(acac)_3_] (Merck, Hohenbrunn, Germany); tetraethyl orthosilicate, TEOS (Merck); polyvinyl alcohol, PVA, with molecular mass of Mw= 145,000 (Merck); methanol, CH_3_OH (Chimopar, Bucharest, Romania); nitric acid, HNO_3_ (Merck); and, distilled water. The reactants were used as received. While using the sol-gel method, starting from the reactant mixture mole ratio of TEOS:H_2_O:PVA:Fe(acac)_3_:MeOH:HNO_3_ = 1:10:(0.63 × 10^−5^ ):0.20:18:0.01, after drying, inorganic-organic hybrid xerogel nanocomposite was obtained. The xerogel material was calcinated in air, for 3 h at several temperatures: 120 °C (P-120), 220 °C (P-220), 260 °C (P-260), 300 °C (P-300), and 400 °C (P-400) with a heating rate of 5 °C/min. 

### 3.2. Characterization

The surface area and porosity were determined with a Quantachrome NOVA 1200e apparatus (Quantachrome Instruments, Boynton Beach, FL, USA) in order to evaluate the morpho-textural properties. The samples were degassed in vacuum for 4 h at room temperature prior to the nitrogen adsorption measurements conducted at 77 K. 

Specific surface areas of samples were calculated by the multiBET method (0.05 to 0.3 P/P_0_) and Langmuir method (0.05 to 0.3 P/P_0_). By only using the V-t method, the micropore surface area was determined (in the linear range, 0.15 to 0.75 P/P_0_). The total pore volume was determined at the last point of adsorption for each sample. The micropore volume was calculated by the Alpha-S method. The pore sizes were determined by both the BJH (Barrett-Joyner-Halenda) and the DFT methods. The BJH method (adsorption) indicates the mesoporosity size of pores and the DFT (equilibrium) indicate the microporosity and mesoporosity. The pore size distribution was calculated while using the NLDFT equilibrium model with cylindrical pores.

Thermal analysis was made with an 851-LF 1100-Mettler Toledo apparatus (Mettler-Toledo GmbH, Bucharest, Romania) in the temperature range 25 and 800 °C in air flow at 5 °C/min. heating rate.

Small Angle Neutron Scattering (SANS) measurements were performed to follow the evolution of the nano- and microstructure of the hybrid nanocomposites versus calcination temperature. The measurements have been completed on the Yellow Submarine pin-hole type SANS instrument located at the Budapest Research Reactor. After the standard normalization procedure, the recorded intensity was represented versus the Q scattering vector ($Q=\frac{4\pi }{\lambda }sin\left(\frac{\theta }{2}\right),$ where λ is the used neutron wavelength and θ is the scattering angle). The used Q range was: 0.01 Å^−1^–0.3 Å^−1^. 

X-Ray diffraction (XRD) was performed while using a Panalytical X’Pert Pro MPD diffractometer using a Cu X-ray (λ = 1.5406 Å) tube and an X’Celerator detector. The resulted patterns were compared with JCPDS database and the Scherrer equation determined the crystallite medium size.

The samples for Transmission Electron Microscopy (TEM) investigation were supported onto meshes holey carbon coated copper grids (Ted Pella) of 300 nm. A High-Tech HT7700 apparatus executed the image recording and analysis. The crystallite size determination was performed by ImageJ software (Version 1.52, University of Wisconsin, Madison, WI, USA).

A Cordouan Wallis Analyser apparatus was used to determine the zeta potential of the colloidal solution of P-260 particles. The sample was diluted in bi-distilled water and chloroform to 1mg/mL. The measurement temperature was fixed to 25 ± 1 °C, saturated signal at a laser power 35%, applied field (automatic), and resolution (medium, 0.8 Hz). The measurement was repeated five times, and the average from them was reported as the final result.

### 3.3. In vitro Cytotoxicity Testing

The selected superparamagnetic nanocomposite sample was tested for *in vitro* cytotoxicity. The viability and morphology of the cells were evaluated after they have been exposed to the agent (magnetic nanocomposite sample) of cell culture. Two cell lines were considered: HaCaT—the human keratinocyte cell line—and the A375 tumour cell line of human melanoma. The effect of the cell viability was tested with the MTT technique. 

### 3.4. Cell Culture

The cell lines used in this study were HaCaT—immortalized human keratinocytes (ATCC, LGC Standards GmbH, Wesel, Germany) and A375—human melanoma (CRL-1619TM, ATCC, LGC Standards GmbH) acquired as frozen vials. 

The cell lines were grown in Dulbecco’s Modified Eagle’s medium (DMEM) with a glucose concentration of 4.5 g/L to which fetal bovine serum (FBS) at a rate of 10% was added as a supplement and mixture of penicillin/streptomycin antibiotics (1%) was also added. During the experiment, the cells were maintained under standard culture conditions: humidified atmosphere at 37 °C and 5% CO_2_ concentration (Steri-Cycle i160 incubator; Thermo Fisher Scientific, Waltham, MA, USA).

### 3.5. Cell Viability Assessment—MTT [3-(4,5-Dimethylthiazol-2-yl)-2,5-diphenyltetrazolium bromide] Assay

The cell viability rate was evaluated by the means of the MTT technique. The cells were grown in 96-well culture plate at density of 1 × 10^4^ cells/well and allowed to adhere to the bottom of the plate until an appropriate confluence (ranging from 80–85%) was reached and the cells were then stimulated with the nanocomposite at concentrations of 10 μg/mL, 30 μg/mL, 50 μg/mL, 100 μg/mL, and 200 μg/mL, for 24 h. At the end of the stimulation period, 10 µL of tetrazolium salt solution was added in each well and the plate was incubated for 3 h. In this period, the mitochondrial reductase of metabolically active cells transforms the yellow tetrazolium salt into a dark blue insoluble formazan compound. The insoluble crystals were further solubilized with 100 µL/well solubilization buffer (mixture of 10% SDS and 0.01 M HCl) and the optical density of each well was spectrophotometrically determined by measuring the absorbance of the samples at 570 nm wavelength while using a microplate reader (xMarkTM Microplate, Bio-Rad Laboratories, Inc. Life Science Research Group 2000 Alfred Nobel Drive Hercules, CA, USA). 

### 3.6. Migratory and Proliferative Potential – aA Wound Healing Technique

The *in vitro* scratch assay method was employed to evaluate the migratory potential of HaCaT cells after treatment with five different concentrations (10 μg/mL, 30 μg/mL, 50 μg/mL, 100 μg/mL, and 200 μg/mL) of NC sample dissolved in DMSO. This technique is considered to be an economical, facile, and quick method to assess cell-to-cell interactions [70]. A number of 2 × 10^5^ cells/well were seeded onto a 12-well plate and they were allowed to reach 85–90% confluency. In the next step, a line (scratch) was drawn in the middle of each well with a sterile pipette tip of 10 μL. All of the cellular debris and detached cells that resulted from the line drawing were removed by washing each well with 1.5 mL PBS. Further, the artificial scratches were exposed to five different concentrations (10 μg/mL, 30 μg/mL, 50 μg/mL, 100 μg/mL, and 200 μg/mL) of NC sample dissolved in DMSO and the cell monolayer recovery was observed by taking pictures of the scratched line at 0 h, 3 h, and 24 h. The images were captured at 10 times magnification while using an inverted microscope Optika Microscopes Optikam Pro Cool 5 and Optika View software (Optika Italy, Ponteranica, Italy).

## 4. Conclusions

Iron oxide–silica nanocomposite containing 20% iron oxide particles has been synthetized and thermally treated at different temperatures. The materials demonstrated superparamagnetic behaviour and they were further characterized for morpho-textural and toxicological properties. The thermal treatment temperature governed the silica matrix texture and iron oxide type and phase. The silica-iron oxide composite material thermal stability and the temperatures for iron oxide crystalline phase formation were determined by thermal analysis. The crystalline phase formation started at lower temperatures (around 225 °C) than usual, and the pure maghemite phase is present up to 275 °C when the hematite transformation is starting. Parabolic evolution of the specific surface areas and micropore/mesopore ratios were observed with a maximum at the 300 °C thermal treatment temperature. The crystallite sizes that were determined from TEM, SANS, and XRD were in good agreement. By increasing the thermal treatment temperature, a slow crystallite growth was observed, while, from 300 °C to 400 °C, thermal treatment temperature the crystallites size increased almost 2.5 times, which was possibly due to the cluster formation of a few iron oxide crystallites. The polycrystalline character of the composites was present at all of the thermal treatment temperatures. Model analysis of SANS data showed that the crystallites were already present at 260 °C, therefore the sample that was calcined at 260 °C has been chosen to be subjected to further *in vitro* assessments.

The *in vitro* cytotoxicity results revealed that high concentrations (100, 200 μg/mL) of P-260 sample solubilized in PBS did not induce an inhibitory effect on HaCaT cell viability, whereas the same concentrations of P-260 sample solubilized in DMSO induced a decrease in HaCaT cell population. For this reason, PBS is recommended for further *in vitro/in vivo* experiments.

The *in vitro* migratory capacity of the healthy keratinocytes (HaCaT) was not affected by low concentrations (10 μg/mL, 30 μg/mL) of the P-260 sample solubilized in DMSO. These concentrations of P-260 sample solubilized in DMSO that induced a decrease in HaCaT cell population are much higher than the concentrations that were used in phantoms for MRI, therefore their use is considered to be safe for this application.

## Figures and Tables

**Figure 1 molecules-25-00653-f001:**
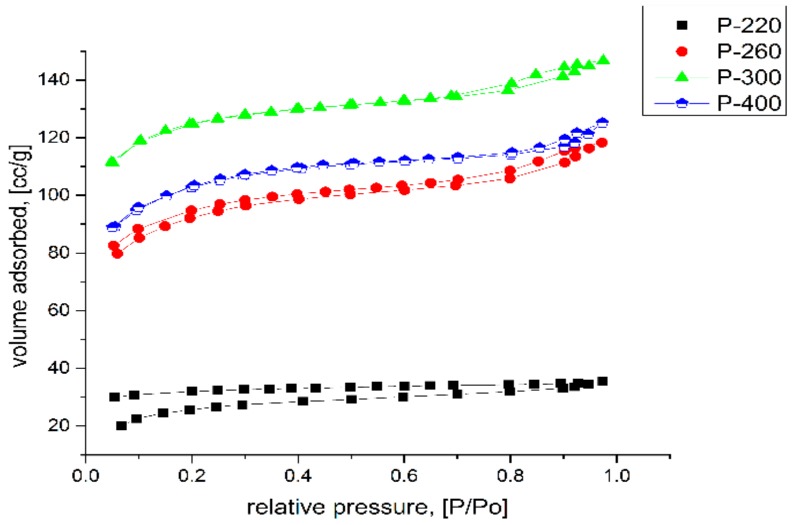
Nitrogen adsorption-desorption isotherms of calcined nanocomposite samples.

**Figure 2 molecules-25-00653-f002:**
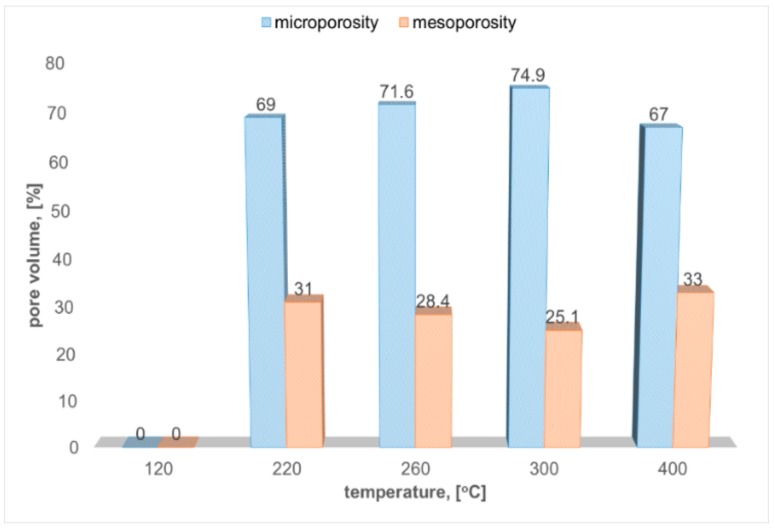
Microporosity/mesoporosity evolution with firing temperature.

**Figure 3 molecules-25-00653-f003:**
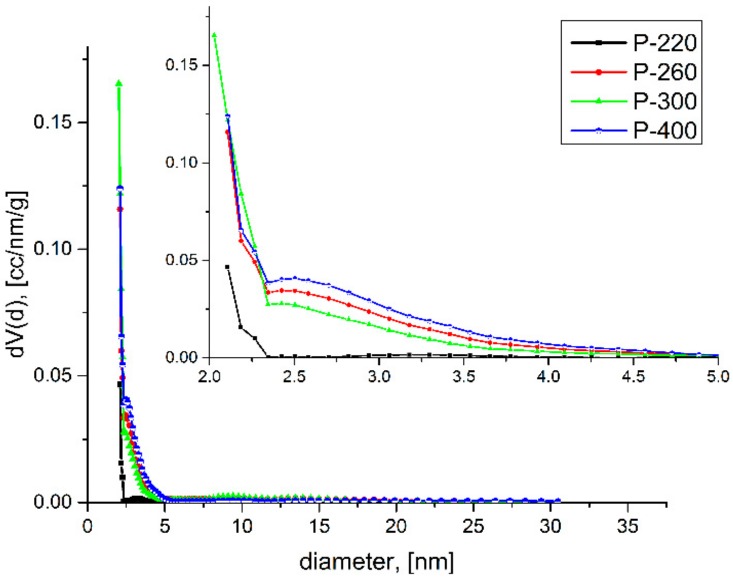
Pore size distribution evolution with firing temperature.

**Figure 4 molecules-25-00653-f004:**
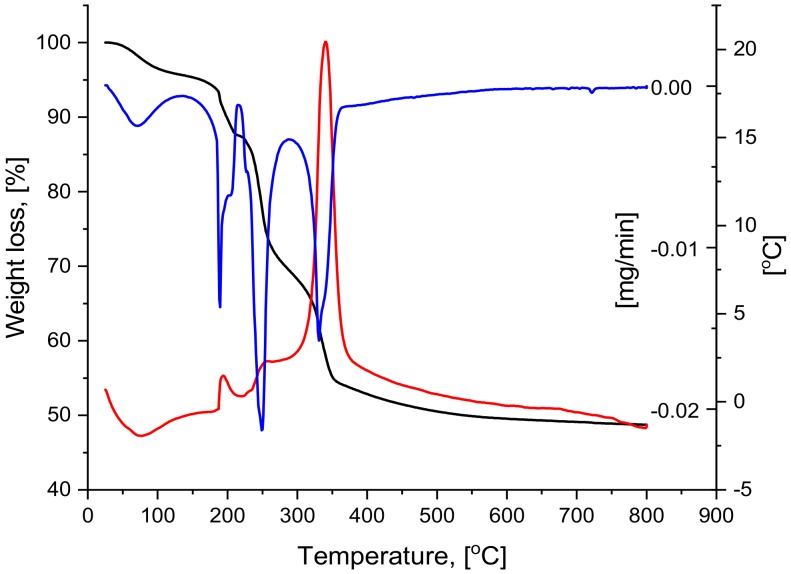
Thermal analysis (TG/DTG/DTA) for silica-iron xerogel composite material.

**Figure 5 molecules-25-00653-f005:**
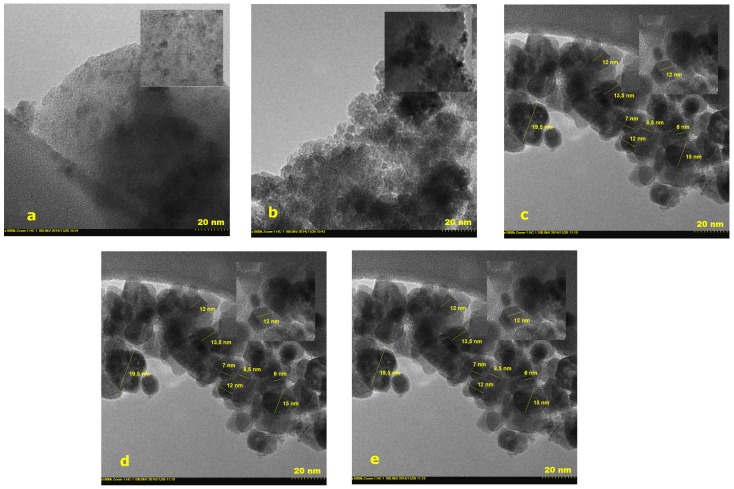
Transmission electron microscopy (TEM) recorded at different thermal treatment temperatures (**a**) 120 °C, (**b**) 220 °C, (**c**) 260 °C, (**d**) 300 °C, and (**e**) 400 °C.

**Figure 6 molecules-25-00653-f006:**
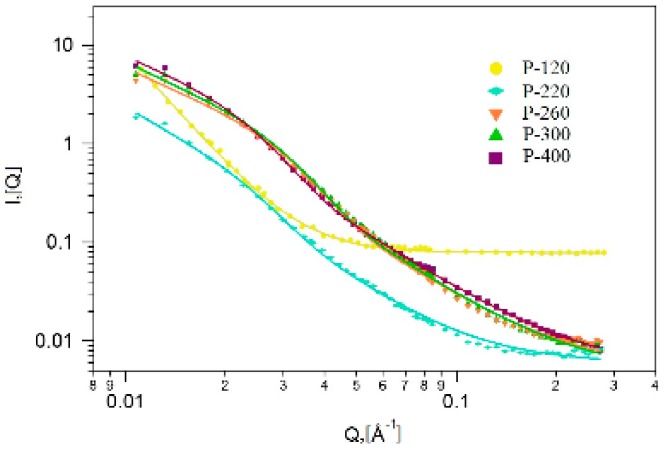
Measured and modelled SANS intensities versus scattering vector of hybrid xerogels calcined at different temperatures.

**Figure 7 molecules-25-00653-f007:**
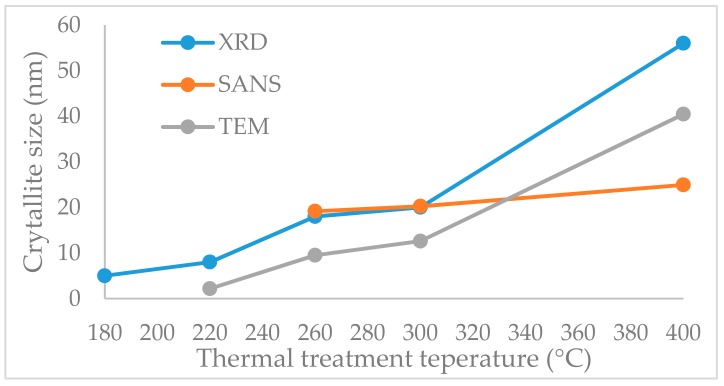
Evolution of the crystallite size for the samples obtained by thermal treatment at different temperatures.

**Figure 8 molecules-25-00653-f008:**
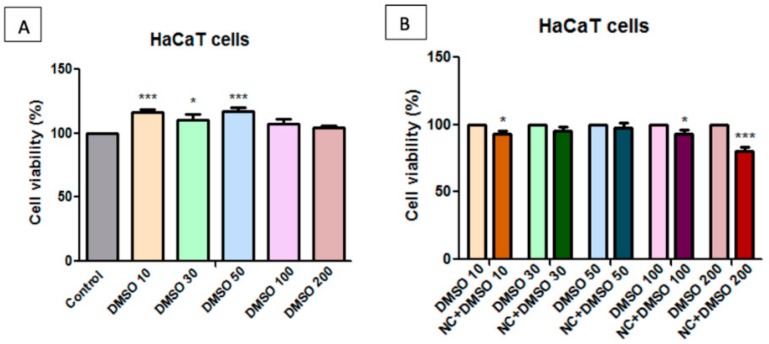
**A**—Effect of dimethyl sulfoxid (dmso) concentrations on human keratinocytes (hacat) cell viability at 24 h post-stimulation. the results are presented as cell viability rate (%) normalized to control cells (no stimulation); **B**—the effect of different concentrations of nanocomposite (nc) samples on hacat cell viability. results are presented as cell viability rate (%) normalized to solvent (dmso) treated cells. to determine the statistical differences one-way anova was employed, followed by bonferroni post-test (* *p* < 0.05; *** *p* < 0.001).

**Figure 9 molecules-25-00653-f009:**
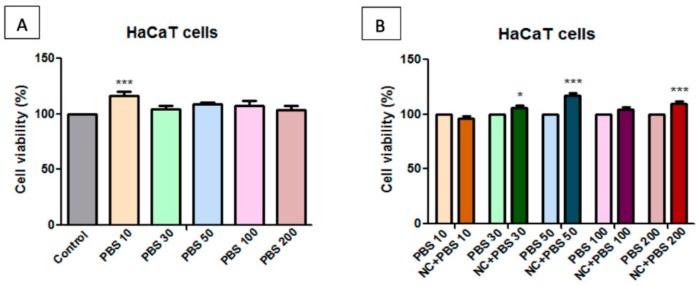
**A**—Effect of phosphate saline buffer (PBS) at different concentrations on human keratinocytes – HaCaT cell viability at 24 h post-stimulation. Results are presented as cell viability rate (%) normalized to control cells (no stimulation); **B**—The effect of different concentrations of NC sample on HaCaT cell viability. Results are presented as cell viability rate (%) normalized to solvent (PBS) treated cells. To determine the statistical differences One-way ANOVA was employed, followed by Bonferroni post-test (* *p* < 0.05; *** *p* < 0.001).

**Figure 10 molecules-25-00653-f010:**
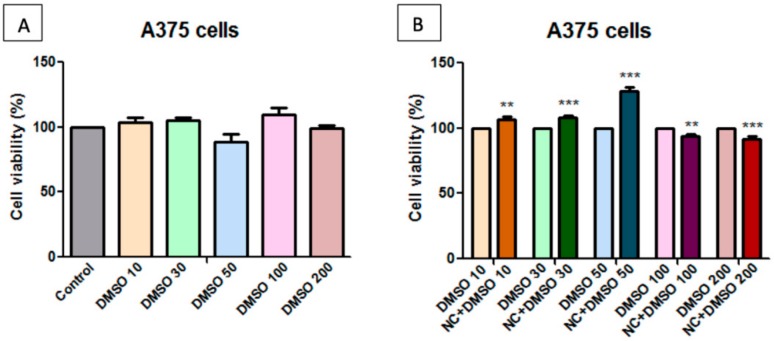
**A**—Effect of DMSO concentrations on human melanoma—A375 cell viability at 24 h post-stimulation. The results are presented as cell viability rate (%) normalized to control cells (no stimulation); **B**—The effect of different concentrations of NC sample on A375 cell viability. Results are presented as cell viability rate (%) normalized to solvent (DMSO) treated cells. To determine the statistical differences One-way ANOVA was employed, followed by Bonferroni post-test (** *p* < 0.01; *** *p* < 0.001).

**Figure 11 molecules-25-00653-f011:**
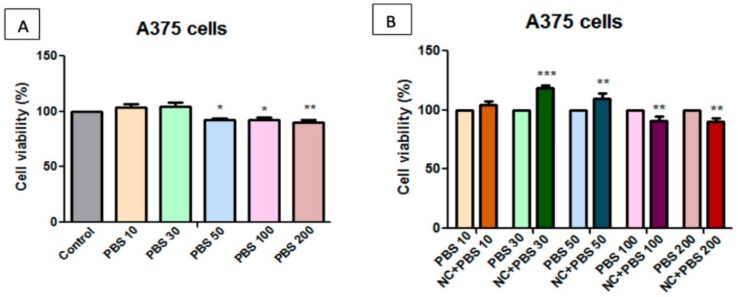
**A**—Effect of PBS at different concentrations on human melanoma—A375 cell viability at 24 h post-stimulation. Results are presented as cell viability rate (%) normalized to control cells (no stimulation); **B**—The effect of different concentrations of NC sample on A375 cell viability. Results are presented as cell viability rate (%) to solvent (PBS) treated cells. To determine the statistical differences One-way ANOVA was employed, followed by Bonferroni post-test (* *p* < 0.05; ** *p* < 0.01; *** *p* < 0.001).

**Figure 12 molecules-25-00653-f012:**
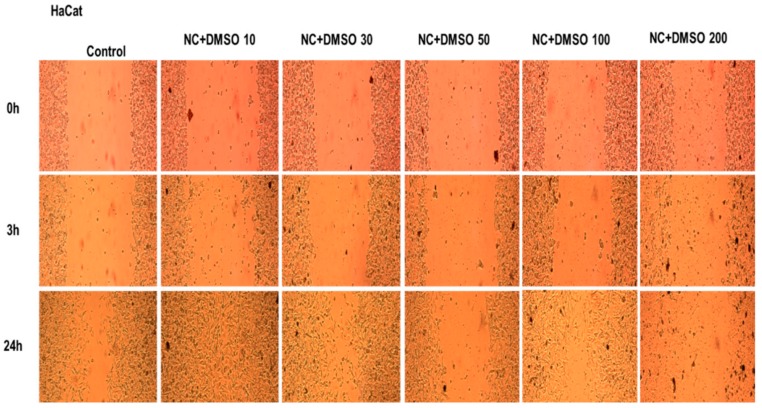
The effect of different concentrations of NC solubilized in DMSO on human keratinocyte (HaCaT) migration and proliferation.

**Table 1 molecules-25-00653-t001:** Morpho-textural parameters of calcined samples.

Samp.	S_BET_^*1^ [m^2^/g]	S L^*2^ [m^2^/g]	S t-plot^*3^ [m^2^/g]	d_DFT*_^4^ [nm]	d_BJH_^*5^ [nm]	TPV ^*6^ [cm^3^/g]	Alpha-S^*7^ [cm^3^/g]	d_fs_ FHH ads^*8^	d_fs_ FHH des^*8^
**P-120**	N/A	N/A	N/A	N/A	N/A	N/A	N/A	N/A	N/A
**P-220**	99	158	64	2.7	3.7	0.055	0.038	2.69/2.89	2.77/2.92
**P-260**	344	519	246	2.6	3.1	0.183	0.131	2.77/2.92	2.57/2.85
**P-300**	468	640	349	2.1	3.4	0.227	0.170	2.85/2.95	2.68/2.89
**P-400**	382	539	258	2.4	3.1	0.194	0.130	2.82/2.94	2.55/2.85

*^1^ – Surface area (multiBET), *^2^ – Surface area (Langmuir). *^3^ – Surface area of micropores (t-plot), *^4^ – Pore width (Density Functional Theory, DFT), *^5^ – Pore size distribution (BJH). *^6^ – Total pore volume. *^7^ – Micropore volume (alpha-S), *^8^ –Surface Fractal Dimension (Frenkel-Halsey-Hill, FHH).

**Table 2 molecules-25-00653-t002:** Weight losses, thermal effects and decomposition domains of the silica-iron oxide nanocomposite.

ΔT, [°C]	Tmax, [°C]	Weight Loss			Thermic Effect
		ΔG/ΔT [%]	Cumulativ ΔG, [%]	ΔG/ΔT [mg]	
25–125	71	4.16	4.16	0.83	endothermic
125–215	193	8.27	12.43	1.65	exothermic
215–275	257	16.69	29.27	3.34	exothermic
275–800	361	22.31	51.44	4.46	exothermic

**Table 3 molecules-25-00653-t003:** Parameters from the SANS curves modelling.

Sample Name	Calcination Temperature	*Rg* (nm)	*p*	Fractal Dimension
P-120	120 °C	-	3.82 ± 0.03	D_surface_ = 2.18
P-220	220 °C	10.01 ± 0.09	2.42 ± 0.01	D_volume_ = 2.42
P-260	260 °C	7.54 ± 0.02	2.21 ± 0.01	D_volume_ = 2.21
P-300	300 °C	7.79 ± 0.02	2.26 ± 0.01	D_volume_ = 2.26
P-400	400 °C	9.59 ± 0.03	2.18 ± 0.01	D_volume_ =2.18

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
