# Peer review of "SiO2-PVA-Fe(acac)3 Hybrid Based Superparamagnetic Nanocomposites for Nanomedicine: Morpho-textural Evaluation and In Vitro Cytotoxicity Assay"

_molecules, 2020, doi:10.3390/molecules25030653_

Round 1
Reviewer 1 Report
The manuscript submitted by Zoltán Dudás reports a strategy for synthesis of Fe2O3/SiO2 composites from the Fe(acac)3/silica/PVA xerogel obtained by one pot acid catalysed sol-gel synthesis using the mixture of iron(III) acetylacetonate, tetraethylorthosilicate (TEOS) as silica source, and polyvinyl alcohol (PVA) with molecular mass of Mw= 145000. The final iron oxide silica composites were obtained by the calcination of xerogel in air at different temperatures. The composites were characterized by the nitrogen porosimetry, XRD, small angle neutron scattering (SANS) and transmission electron microscope (TEM). Additionally, in vitro cytotoxicity investigation was conducted for the selected composite sample. In my opinion the topic is very interesting and the presented results are sound. However, the article should additionally be supplemented with more precise description and discussion of the presented data. I think, the paper needs major revision before consideration for publication in Molecules. There are several major questions that affect the final conclusions.
The English should certainly be improved throughout the paper. It should be looked through and corrected by a native English speaker. I have already found a few grammar (e.g. an inappropriate use of tenses) and spelling errors in the text, though I'm not the native English speaker.Moreover, there are a plenty of awkward constructs e.g.:
Line 118-119: The thermally induced textural modifications, of the P260 sample, were underlined using nitrogen porosimetry.
Line 192: Between 180 °C and 300 °C a slow but constant,…
Line 200: From 220 ° C on the crystal formation and growths are clearly visible.
Line 419: Prior analysing in nitrogen environment, …
Etc…
The Introduction should be improved, especially when it comes to the goal of the paper. The purpose of the work is shortly mentioned only in the Abstract, but it should be clearly defined in the Introduction. The synthesis and the characterization of samples cannot be a goal of the scientific article. Lines 68-70: The quotation [26] is inappropriate. From Ref. 26 it follows that the concept of “cell type dependent nanotoxicity” was introduced by Blechinger, J. et al., thus the paper Blechinger, J. et al. Uptake kinetics and nanotoxicity of silica nanoparticles are cell type dependent. Small 9, 3970–3980, 3906 (2013). https://onlinelibrary.wiley.com/doi/full/10.1002/smll.201301004 should be quoted. Lines 80-82: Repetition in the text Lines 79, 89, 92, 94: The style of quotation is different from the Molecules style; e.g. Elias A et al. (2009) ; Malvindi, M.A. et al. 2011, etc. It should be improved. Nitrogen porosimetry: In my opinion the discussion concerning the porosity of samples should be significantly improved. Firstly, I cannot agree that nitrogen porosimetry give information about the thermally induced textural modifications. From N2 adsorption/desorption measurements the parameters characterizing the porosity of samples are obtained. From the shape of isotherms the possible shape and size of pores can only be deduced.Authors should take into account that the internal structure of composites is disordered (as it follows from TEM micrographs). Moreover, the adsorption and desorption branches of isotherms do not overlap; not only at the low relative pressure ranges but in fact in the whole p/p0 range. The hysteresis loops are hardly visible. Thus, discussion concerning the presence of micro- and macropores should be more cautious. Similarly, the BJH and DFT models should be use with caution. In fact, there is lack of discussion concerning the non-overlaping branches of isotherms.
Besides, there is no information concerning the conditions of N2 adsorption; i.e. no information if the N2 adsorption was measured at pressures which allow to estimate the microporosity of samples.
Line 129: What does it mean: “At 260 °C the porosity increased.” To this statement the explanation and discussion should be added; e.g. from the synthesis procedure and from http://www.doiserbia.nb.rs/img/doi/1820-6131/2016/1820-61311604265I.pdf, it follows that at that temperature most of the organic phase and precursors decomposed. Thus, the free volumes appear in the samples calcined above 260 deg.
Lines 134-136: From http://www.doiserbia.nb.rs/img/doi/1820-6131/2016/1820-61311604265I.pdf, it follows that at 120 deg. the organic phase is still present in the xerogel. Also, the presence of undecomposed precursors can be expected. Thus, the xerogel calcined at 120 deg. exhibits low application potential thus discussion concerning its porosity can be omitted.
Fig. 2: One should take into account the experimental error in the measurement of SBET (it is in limit ±7%), in the case of Vp is in limit ±10% and in the case of Dp is in limit ±6% (V.A. Hackley, A.B. Stefaniak, Real-world precision, bias, and between-laboratory variation for surface area measurement of a titanium dioxide nanomaterial in powder form, J. Nanopart. Res. 15 (2013) 1742). Thus, the discussion of changes in the porosity between the samples should be more careful.
Fig. 3: The title of the Y title is incorrect; There should be the dV/dD or dV/dlog(D). There is lack information how was the PSD (presented in Fig. 3) calculated. Were the adsorption or desorption branches used for PSD?
Thermal analysis of xerogel should be added. TG and DSC should facilitate the explanation and understanding the changes of the porosity of composites with regard to the presence of organic moieties from precursors and PVA. Line 192: Please explain the meaning "a slow but constant, almost linear increase of the crystallites". Such statement is unjustifiable since there is no measurement of the kinetics of the crystallites growth process. In my opinion, the XRD diffractograms of composites should be added with the assignment of the characteristic peaks. It is known that Fe2O3 is sensitive to the temperature, and such phenomena as crystallite growth and changes in the phase composition occur. There is no substantive discussion that gives insight into these processes. Without XRD, the statement "Up to 300 °C only cubic maghemite phase could be determined, while at 400 °C the hematite phase appeared in addition." cannot be verified. The XRD peaks (in fact diffraction reflexes) should be given on the basis of which the crystallite sizes were determined (presented in Fig. 5). Discussion concerning the estimation of the crystallite size from the Scherrer equation should be conducted taking into account the limitations of the method. For the statement; "The crystallite sizes are in good correlation with the sizes determined from the XRD and SANS measurements." the calculations on the basis of TEM micrographs should be presented. In the Fig. 5 the Fe2O3 phase is not indicated. Where is the SiO2 phase? In Table 2, the names of samples appear. They are not given in other part of the manuscript and are not defined in the 3.1. paragraph. Similarly, in line 364 the abbreviation NC is given. Please define the names of samples in the 3.1 and use them consequently in the article. In my opinion discussion concerning the SANS and cytotoxicity are very valuable. 3.1. Materials and synthesis protocol:This Section should include the detailed description of the conducted research. In my opinion, details concerning preparation of the xerogel and composites should be added. Especially, details concerning the rate of the calcination should be given since it is known that there a significant impact of mass transport effects in calcination of xerogels.
3.2Why degassing was conducted at room temperature?
How was determined that degassing procedure i.e. for 4 hours at room temperature is appropriate for the investigated samples. The p/p0 from which the nitrogen adsorption was measured should be given since the microporosity of samples is discussed.
Line 424: The formula for Q calculation (in the bracket) should be checked if it is correct. Line 445, 464 & title: The paper should be edited and thoroughly checked taking into account superscripts and subscripts, e.g. SiO2≠SiO2 2x105 cells (?) ect.Author Response
Please see the attachment.

Reviewer 2 Report
The manuscript entitled “SiO2-PVA-Fe(acac)3 hybrid based superparamagnetic nanocomposites for nanomedicine: Morpho-textural evaluation and the In Vitro cytotoxicity assay” reports the characterization of a previously reported nanocomposite intended to be used in magnetic resonance imaging in terms of temperature-induced morpho-textural modifications using small angle neutron scattering and nitrogen porosimetry. Additionally, the induced cell toxicity by the nanocomposite was studied using MTT assay in human keratinocytes and human melanoma cells.
According to this reviewer, this manuscript fails to report a significant advance on the characterization of MRI materials. The nanocomposites presented were previously reported by some of the authors (Curr. Org. Chem. 2017, 21(27), 2783-2791, 646 https://doi.org/10.2174/1385272821666170428123451.) Only additional characterization in terms of structure and porosity analysis were included. as the authors present the synthesis of the materials as part of the new experimental work, many characterization are missing (including, but not only, XRD which is commented in the text, but not showed) such as VSM to demonstrate the superparamagnetic property, EDS coupled to the TEM to infer chemical composition and DLS and z-potential to demonstrate stability in solution (essential to understand the results of toxicity tests).
The other part of the study is based on the cell toxicity presented. However, the cells lines used and one of the tests have limited relevance for the study of biocompatibility of nanocomposites intended for MRI. Nor keratinocytes either melanocytes are the most relevant models for this kind of study due to the fact that MRI is not the preferred diagnostic technique used for melanoma or skin cancer and the administration route is nor dermal. Additionally, the migration and proliferation study does not provide enough significant further insight. An acceptable approach could have been studying the impact in toxicity of the different nanocomposites generated at different temperatures, but using other cell models.
Taking into account the stated above among other considerations, according to this reviewer, this manuscript should not be considered further for publication in Molecules.
Round 2
Reviewer 1 Report
The manuscript submitted by Professors by Zoltán Dudás has been significantly improved. The Introduction has been improved especially in terms of the goal of the paper. The purpose of the work has been clearly defined. Thermal analysis of xerogel as well as the detailed description of the conducted research have been added.
This manuscript is acceptable now. I recommend to publish the work in Molecules.
Reviewer 2 Report
The authors made a significant an appreciated effort to improve the manuscript in terms of new data about characterization and convinced this referee about the adequacy of the cell lines. However, as the authors present the synthesis of the materials as part of the new experimental work, some characterization are missing. It is not clear that their previous work contains VSM to demonstrate the superparamagnetic property.
On the other hand, in order to present significant results for the in vitro cytotoxicity it is necessary to demonstrate their dispersion and stability in solution, so, DLS and z-potential (or equivalent) is still essential to understand the results of toxicity tests.
